# Efficacy of *Bacillus thuringiensis* Treatment on Aedes Population Using Different Applications at High-Rise Buildings

**DOI:** 10.3390/tropicalmed5020067

**Published:** 2020-05-01

**Authors:** Zuhainy Ahmad Zaki, Nazri Che Dom, Ibrahim Ahmed Alhothily

**Affiliations:** 1Center of Environmental Health & Safety, Faculty of Health Sciences, Universiti Teknologi MARA, Puncak Alam 42300, Selangor, Malaysia; zuhainy@gmail.com; 2Integrated Mosquito Research Group (IMeRGe), Faculty of Health Sciences, Universiti Teknologi MARA, Puncak Alam 42300, Selangor, Malaysia; iialhothily@gmail.com

**Keywords:** *Bacillus thuringiensis israelensis (Bti)*, high-rise buildings, larviciding applications

## Abstract

*Bacillus thuringiensis israelensis (Bti)* is an effective biological insecticide for killing mosquito larvae. However, choosing the suitable application method for larviciding is critical in increasing its effectiveness. Therefore, this study aimed to determine the effectiveness of *Bti* (VectoBac^®^) WG using various applications at high-rise buildings. Three different applications of *Bti* treatment were applied at three high-rise buildings in Bandar Saujana Putra. The ULV machine is used for Pangsapuri Impian, a mist blower for Pangsapuri Seri Saujana and a pressured sprayer for BSP 21. BSP Skypark does not undergo treatment and acts as a control. The efficacy of *Bti* treatment was measured by analyzing the ovitrap surveillance data collected (POI and MLT) for pre and post-treatment. Post-treatment ovitrap surveillance indicates that the *Aedes* sp. mosquito density was lower than the density at the time of pre-treatment surveillance. Overall, the *Aedes albopictus* species in both an indoor and outdoor environment setting had shown a reduction. The highest *Aedes* sp. density reduction is seen through the use of mist blowers in outdoor settings for *Aedes albopictus*, (%POI reduction = 87.4%; %MLT reduction = 93.8%). The mist blower yielded results that is significantly higher compared to other larviciding applications; the order from greatest to the least was mist blower > pressured sprayer > ULV. It can be concluded that each application produces different degrees of effectiveness in reducing the *Aedes* sp. density in different environmental settings.

## 1. Introduction

Dengue is endemic in Malaysia; it is found mainly in the urban and suburban areas [1]. The dramatic increase in dengue cases reported is due to several factors such as increased urbanization, housing designs, the mobilization and movement of the population, erratic water supplies and ineffective or unsustainable vector control [2,3]. Several strategies have been implemented in an effort to control the proliferation of dengue vectors, either at their immature aquatic stages or at their mature stages. Dengue cases have also been reported in high-rise buildings that offer places of residence in the form of flats, apartments or condominiums [2]. Human activities and poorly maintained sanitation in the surrounding area can trigger the breeding of mosquitoes [3]. High-rise buildings designed with rain gutters that make cleaning almost impossible offer the best breeding conditions for mosquitoes as regions with improper drainage and piping systems show high potential in becoming Aedes habitats [1]. The abundance of these species is influenced by the preference and inherent behaviors in oviposition of the female mosquitoes, as well as other biotic and abiotic factors [4,5]. Environmental factors such as relative humidity, wind and temperature influence the occurrence and density of these species. Anthropogenic changes in the environment will also influence the abundance and distribution of these species [6]. There are a lot of potential breeding sites in high-rise buildings and it is difficult to control by just focusing on a singular control method. The increase in the number of breeding mosquitoes within the region is the main cause of dengue outbreaks [7].

In vector control, the application of insecticides is widely used with the aim to kill and/or reduce the population size of infected mosquitoes whenever cases of dengue are reported. The application of insecticides, especially pyrethroids, is carried out through space spraying in the form of cold or thermal fogging. The main purpose of space spraying is to kill the infected mosquitoes, thereby interrupting the cycle of transmission, and stopping the spread and occurrence of dengue [8]. Other control activity in order to reduce the density of *Aedes* sp. is larviciding. The larvicide agent, which can kill mosquito larvae, includes biological insecticides such as *Bacillus thuringiensis israelensis (Bti)*, a microbial larvicide that is commonly used as a larvicidal agent [9]. *Bacillus thuringiensis israelensis (Bti)* is a Gram-positive bacterium that is used as a larvicide to cause death to *Aedes* larvae. Delta endotoxin from the *Bti* is ingested by larvae and binds to the cells of the epithelium in the mid gut, causing disruption to the osmotic balance within the mid gut’s epithelial cells by forming pores in the cell membrane and lysing the cells [10].

A lot of studies have been done in Malaysia to determine the effectiveness of each dengue control method [11,12]. Synthetic insecticides have been effectively used for the past several decades as a method of vector control. However, the effective use of chemical insecticides has been greatly impeded by several factors such as the development of physiological resistance within the target vector, the causing of environmental pollution resulting in the bio amplification of food chain contamination, and the development of harmful effects on other beneficial non-target animals [13]. Therefore, the need for more effective and environment-friendly control agents has become more urgent and as of the moment, biological agents seem to be the most promising and safest alternative. *Bacillus thuringiensis israelensis (Bti)*, is an effective larviciding agent for *Aedes* sp., and this method is applicable in the mass treatment of dengue outbreak areas [14]. Larviciding using biological insecticides is effective in killing mosquito larvae and it can be conducted regularly to reduce the density of *Aedes* sp. within a region.

Different treatment applications and different formulations are available. The *Bti* comes with different formulations, such as in the form of rice husks, wettable powder, water granules and water tablets. The larvicidal activity can be applied using an ultra-low volume (ULV) machine, a mist blower, a pressured sprayer, and by direct application. The process of choosing the application method of *Bti* treatments is critical in increasing its effectiveness in reducing the density of *Aedes* sp. Therefore, this study aims to understand the best application method of *Bti* treatments for high-rise buildings by comparing different methods of application and studying its impact on mosquito populations and species. This study is expected to contribute to the baseline knowledge of the possible effect of using different applications on *Aedes* sp. in Selangor, Malaysia. 

## 2. Materials and Methods

### 2.1. Description of Study Site

This study was conducted at Bandar Saujana Putra, in the Kuala Langat District of Selangor. Kuala Langat is situated in the southwestern part of Selangor and covers an area of 885 square kilometers, with a population of 260,261 people. In this study, four high-rise residential buildings; Pangsapuri Impian (PI), Pangsapuri Seri Saujana (PSS), BSP 21 (B21), and BSP Skypark (BSP) were selected as study sites. Figure 1 shows the location of these high-rise buildings. These four localities of high-rise apartments had been selected due to a series of dengue outbreak incidences that had been reported by the Vector Control Unit of the Kuala Langat Health Office from 2014 to 2018. PI, PSS and B21 were chosen as the treatment locality, while BSP was selected as the control locality.

All localities were identified as high-rise buildings in sub-urban residential areas, where two of them are considered low-cost apartments (PI and PSS) and the other two are serviced apartments (B21 and BSP). PI and PSS are located side to side. PI is a five-story, low-cost apartment with eighteen blocks whereby each block consists of eighty houses and PSS is a low-cost apartment with five-story blocks consisting of ten blocks and each block has eighty units of houses. B21 consists of ten blocks of serviced apartments with eighteen and twenty-seven story buildings, while BSP is a serviced residence consisting of two blocks of apartments with six hundred and eighty-nine units of houses. Both PI and PSS have been in service for approximately eight years and the overall surroundings seem poorly maintained with unmanaged trash disposal, untidy communal yards and an overgrowth of vegetation. The surrounding area appears unclean with piles of garbage randomly scattered at almost every floor. In contrast, the surrounding environment of B21 and BSP is well kept with planted trees, with the area looking cleaner and well maintained in terms of sanitation with proper waste management systems in place. However, a certain area in B21 was quite messy as the area was still under construction with works progressing in the area.

### 2.2. Study Design

This study applies the field research design to investigate the density of dengue vector mosquitoes within selected high-rise buildings in Kuala Langat, Selangor after applying the biological control treatment using different treatment applications. This study was conducted to identify which treatment application is more effective to be applied in high-rise building areas (Table 1). Positive ovitrap index (POI) and mean larvae per ovitrap (MLT) were used to assess the density of *Aedes* sp. with regard to their distribution and abundance. Then, the treatments were carried out shortly after the baseline study has been completed. This was conducted for about twelve weeks in all the localities. Three different types of applications were applied using *Bti* (VectoBac^®^) WG at each locality over the twelve-week treatment duration. The PI locality was treated by applying *Bti* (VectoBac^®^) WG using an ultra-low volume machine (ULV), PSS was treated using mist blowers (MB) and B21 were treated using a back-pack pressured sprayer (PS). BSP, as a control locality (C), had no treatment applications applied to it. The treatments were conducted at each locality for overall coverage. After each treatment application, post-treatment ovitrap surveillance was conducted to evaluate post-treatment POI and MLT. The post-treatment ovitrap surveillance was conducted in a similar manner to the baseline ovitrap surveillance and it was done shortly after each treatment was completed. The results were used as a measure to compare the effectiveness of different treatment applications and also to compare the results of the baseline ovitrap surveillance with the post ovitrap surveillance after treatment.

### 2.3. Treatment Application

A *Bti* formulation, VectoBac^®^ WG (Lot No. 114-114-3L) was used in this study. The *Bti* treatments were targeted towards all identified actual and potential outdoor breeding areas. Table 2 shows all the equipment used in this experimental field study. The ULV sprayer can cover a wide area whilst the mist blower and pressured sprayer provides a more targeted approach, which is contained to a specific area. The application is direct in the sense that the *Bti* were basically sprayed directly into the potential breeding containers. 

### 2.4. Pre and Post-Treatment Ovitrap Surveillance

The profile of Aedes density was measured through the data collected from conventional ovitrap surveillance. Ovitrapping was done to obtain baseline data of the infestation profile before treatment (pre-treatment) and after treatment (post-treatment) using the same method. The baseline profile of *Aedes* sp. density was measured through a conventional ovitrap surveillance that was conducted from 3 December 2018 to 4 March 2019 (14 weeks). This was done at all four localities to obtain the baseline information of Aedes ovipositions. After that, a series of three treatments were conducted for all localities. The first treatment was initiated on 25 March 2019, two weeks after the pre-treatment surveillance. The second treatment was conducted on 3 April 2019 and the third treatment on 8 May 2019. A post-treatment ovitrap surveillance was conducted directly after each treatment. In this study, the ovitraps were placed at places deemed as “semi indoors” and “outdoors”. Semi indoors would refer to areas inside the building itself, namely areas covered by the roof of the building whilst outdoors refers to areas outside the building area, including the surrounding environment [5]. The ovitraps were placed randomly near potential breeding spots in order to capture more accurate results. Other considerations taken is for them to be placed in areas with less physical and environmental interference in order to reduce the risk of misplaced or malfunctioning ovitraps. The ovitraps were recovered after five days from their designated areas and were brought back to the laboratory for larvae species identification. At every locality, ovitrapping was done during three independent visits. For each visit, a total of a hundred ovitraps were placed. Fifty ovitraps were placed accordingly in semi indoors and outdoors settings and they were distributed randomly up to level five of the buildings.

### 2.5. Measurement of Positive Ovitrap Index (POI) and Mean Larvae per Trap

The density of the *Aedes* sp. mosquito of pre- and post-treatment was measured by calculating the POI and MLT for each different species at different environmental settings. Both recovered and missing ovitraps were recorded. Positive ovitraps with positive mosquito eggs were identified. A comparison of the POI in each environmental setting for each of the four localities was calculated to determine the Aedes profile of the area. The POI was determined by dividing the number of positive ovitraps with the number of recovered ovitraps during collection, multiplied by 100 to obtain a percentage [7].

Mosquito eggs were brought to the laboratory for the hatching process. The number of developed larvae was subsequently counted and these were observed at the third instar to identify the *Aedes* sp. The number of Aedes larvae was calculated and recorded for each positive ovitrap. The identification process is conducted with supervision from entomologists for more accurate results. The identification of larvae was carried out by looking at the comb teeth and siphon characteristics. The MLT was determined by dividing the total number of larvae with the number of recovered ovitraps. The POI and MLT for both semi indoor and outdoor settings were then categorized to reflect information based on the two separate species; *Aedes aegypti* and *Aedes albopictus*. 

### 2.6. Statistical Analysis

The results from ovitrap surveillance between 3 December 2018 to 4 March 2019 were gathered for analysis. Positive ovitrap index (POI) and mean larvae per ovitrap (MLT) were calculated to assess the density of *Aedes* sp. with regard to their distribution and abundance. The post study was conducted in a similar manner to the baseline study shortly after the treatments were completed in order to identify the *Aedes* sp. density based on POI and MLT results at all localities after the treatment. The result was used as a measure to compare the effectiveness of different treatment applications and also to compare the results of the baseline ovitrap surveillance with the post ovitrap surveillance after treatment. All data obtained from this study were analyzed using the Statistical Package for the Social Sciences (SPSS) software version 23. The significant difference was determined by using the Paired sample *t*-test, Analysis of Variance (ANOVA) and hierarchical regression. 

## 3. Results

The summary data of the efficacy of the *Bti* (VectoBac^®^) WG treatment based on POI and MLT were described in Table 3. Table 3A shows the percentage of reduction in POI between pre-treatment and post-treatment surveillance. For semi-indoor settings, the data shows that the *Aedes albopictus* species is clearly affected by the *Bti* treatment for all application methods, whilst the treatment has no effect on the density of *Aedes aegypti* species. In contrast, the data for the outdoor settings show that both species are affected by the *Bti* treatment, regardless of the application method. Table 3B shows the percentage of reduction in MLT between pre-treatment and post-treatment surveillance. In both settings, *Aedes albopictus* shows a reduction in terms of numbers of larvae, while for the semi indoor setting, *Aedes aegypti* only show a reduction in MLT for the treatment application using a pressured sprayer. Generally, the POI and MLT decreased dramatically at all localities, two weeks after initiating *Bti* (VectoBac^®^) WG treatments, especially in the outdoor environmental setting, showing that this kind of control treatment activity is suitable for outdoor environments.

### 3.1. Distribution of Pre- and Post-Treatment for All Applications Based on the Positive Ovitrap Index (POI) of Dengue Vectors in Different Environmental Settings

Figure 2 shows the results of pre- and post-treatment for all applications based on the POI of the *Aedes aegypti* and *Aedes albopictus* species in different environmental settings. The results of a paired-sample *t*-test show that there is no significant difference for both species on the reduction of POI using *Bti* treatments for all applications in semi-indoor settings (Figure 2A,B). In contrast, the results of the paired *t*-test for outdoor settings show a significant difference between pre- and post-treatment. The application method of the pressure sprayer recorded a significant reduction of POI with POI*_pre_* = 5.4% and POI*_post_* = 2.0%; (*p* = 0.02) for the *Aedes aegypti* species, while the mist blower and ULV applications both show a reduction of POI with POI*_pre_* = 31.7% and POI*_post_* = 4.0%; (*p* = 0.01) and POI*_pre_* = 36.3% and POI*_post_* = 7.9%; (*p* = 0.02) respectively, for the *Aedes albopictus* species in the outdoor environment for high-rise buildings (Figure 2C,D). Therefore, it can be concluded that there is a reduction of POI for both species in outdoor settings after treatment with *Bti* (VectoBac^®^) WG by using the mist blower and the ULV. This application is suitable for *Bti* treatment in controlling larvae in outdoor environmental settings seeing as the *Aedes albopictus* species recorded the highest reduction during pre- and post-treatment in both environmental settings, especially in the outdoor setting.

### 3.2. Abundance of Pre- and Post-Treatment for All Applications Based on Mean Larvae per Trap (MLT) of Dengue Vectors in Different Environmental Settings

The paired sample *t*-test was also used to examine the effect of *Bti* (VectoBac^®^) WG treatment on the density of the mosquitoes through various treatment applications. The results show that there is no significant difference in terms of mean larvae per trap pre- and post-treatment in both species, for all application methods in semi-indoor settings. Nonetheless, the pressured spray method recorded a significant reduction in MLT for *Aedes albopictus* in semi-indoor settings with MLT*_pre_* = 2.3 larvae/trap and MLT*_post_* = 0.9 larvae/trap; (*p* = 0.03) (Figure 3A,B). The results show a significant difference in the reduction of density for both species pre- and post-treatment in the outdoor setting. Larviciding using the ULV and the pressured spray were found to create a significant reduction in MLT for *Aedes aegypti* (ULV: MLT*_pre_* = 2.1 larvae/trap and MLT*_post_* = 0.7 larvae/trap; (*p* = 0.01); pressured spray: MLT*_pre_* = 1.8 larvae/trap and MLT*_post_* = 0.5 larvae/trap; (*p* = 0.01)). The results obtained showed that larviciding using *Bti* (VectoBac^®^) WG treatments can reduce the density of *Aedes albopictus* using all application methods (ULV: MLT*_pre_* = 14.8 larvae/trap and MLT*_post_* = 1.1 larvae/trap; (*p* = 0.01); mist blower: MLT*_pre_* = 11.2 larvae/trap and MLT*_post_* = 0.7 larvae/trap; (*p* = 0.01) and pressured spray: MLT*_pre_* = 3.9 larvae/trap and MLT*_post_* = 0.6 larvae/trap; (*p* = 0.02) (Figure 3C,D). 

### 3.3. Efficacy of BTI (Vectobac^®^) WG Treatment on Different Application Methods at High-Rise Buildings

The results show that each application method has a different level of effectiveness in reducing the density of *Aedes* sp. Therefore, a hierarchical multiple regression test was used to determine the best treatment application that is suitable and effective in reducing *Aedes* sp. density within the study area. For this study, control locality was used as the controlled variable. To determine whether the controlling effects are significant, a three-step hierarchical regression was conducted. Firstly, the effect of independent variables for larviciding application, namely ULV, mist blower and pressured sprayer, was estimated. Then, the controlling variable (control) was included to measure whether the controlling variables have a significant and direct impact on the dependent variables (POI and MLT). Lastly, the interaction terms between independent variables and moderating variables were entered to show the additional variance. The result was tabulated in Table 4. In summary, the mist blower yielded results that were significantly higher when compared to other larviciding applications; the order from greatest to the least was mist blower > pressured sprayer > ULV. This reflects the effectiveness of these applications for larviciding using *Bti* (VectoBac^®^) WG treatments in outdoor environments. 

## 4. Discussion

The control of dengue vectors is already discussed by several authors [8,10,14,15]. The results obtained in this study recommended that the *Bti* (VectoBac^®^) WG treatment can be conducted using various application methods to reduce the density of mosquito larvae in high-rise buildings, especially for outdoor environments. Many methods are available for the control of dengue vectors, for example, environmental control, chemical control, biological control, genetic control, human-behavioral control and others [16]. The results obtained show the implication of using different applications of *Bti* (VectoBac^®^) WG. None of the studies prior have looked into the efficacy of different application methods; it is important in choosing the right application to be used for control activities. Post-treatment ovitrap surveillance indicates that the *Aedes* sp. mosquito density was lower than the density at the time of pre-treatment surveillance. Overall, the *Aedes albopictus* species in both indoor and outdoor environment settings had shown a reduction in numbers after being treated with *Bti* (VectoBac^®^) WG. The highest *Aedes* sp. density reduction is seen through the use of mist blowers as the treatment application method in outdoor settings for *Aedes albopictus*, where both POI and MLT shows a higher degree of effect after treatment (%POI reduction = 87.4%; %MLT reduction = 93.8%). These results show that the mist blower is the most suitable treatment application for larviciding in high-rise building areas. The dispersal range of a mist blower allows for a focus on targeted areas, making it much better when compared to treatment using ULV. These biological insecticides are proven to be effective in killing *Aedes* sp. larvae [17,18,19,20]. The results show overall effectiveness in reducing *Aedes* sp. density in the study area through larviciding activities done by using this biological treatment. Larviciding using the mist blower is found to be the best treatment application for outdoor environmental settings whilst for semi indoor settings, the pressured sprayer is found to be a more suitable treatment application in killing *Aedes albopictus* larvae under field conditions.

The effectiveness of *Bti* (VectoBac^®^) WG in controlling dengue vectors had been extensively demonstrated worldwide. Indeed, choosing the right application method will increase the effectiveness of control activities. The efficacy of *Bti* treatment using mist blower directly to the breeding site is higher than ULV spraying. The study showed a significant difference in the POI value between the control and the treated high-rise buildings, where the percentage of reduction in POI in the treated high-rise building was higher than the controlled locality. This showed that *Bti* (VectoBac^®^) is proven to be effective in killing *Aedes* sp. larvae and in controlling the *Aedes* sp. population in high-rise buildings. Under field conditions, larviciding using a mist blower was found to be the best application method for outdoor environmental settings whilst for semi indoor settings, the pressured sprayer was found to be more suitable in killing *Aedes albopictus* larvae. In a nutshell, the most effective treatment application was found to be by using the mist blower (*p* = 0.004). This form of application method is found to be effective in controlling *Aedes albopictus* in an outdoor environment due to the ability to produced small droplets that can remain airborne for a sufficiently long time in the treated area to give a knock-down effect to the targeted flying insects. This is an important factor in protecting the environment, especially in terms of avoiding any adverse impact on insect pollinators. Additionally, the mist blower machine could be carried up by personnel and the dispersal is focused toward targeted areas. This application method is said to be able to overcome the problem faced by local authorities and health department officers in doing full-coverage of vector control activities. However, each application method produces different degrees of effectiveness in reducing both species of *Aedes* sp. in different environmental settings. Thus, the right application method based on housing type and environmental setting is necessary for the effective reduction of the *Aedes* sp. population.

## 5. Conclusions

The application of *Bti* (VectoBac^®^) WG in vector control management is relevant and serves as an effective tool to control the *Aedes* sp. population. It is recommended to be conducted regularly. Nevertheless, it is important to choose the right application method in order to make sure the effectiveness of control activities. On the contrary, the long-term solution to the dengue scourge is proper maintenance and cleanliness. The reduction of larval breeding sites is preferred as one of the most effective strategies. This method combined with other control methods may present a strong strategy in reducing the *Aedes* sp. population in the long run. The findings from this study will help to identify which application method is most effective for *Bti* treatment in high-rise buildings. Hence, it is recommended that the *Bti* (VectoBac^®^) WG treatment be conducted using various application methods to reduce the density of mosquito larvae in high-rise buildings, especially for outdoor environments. Therefore, hopefully, this study can act as a guideline for future control activities in high-rise buildings.

## Figures and Tables

**Figure 1 tropicalmed-05-00067-f001:**
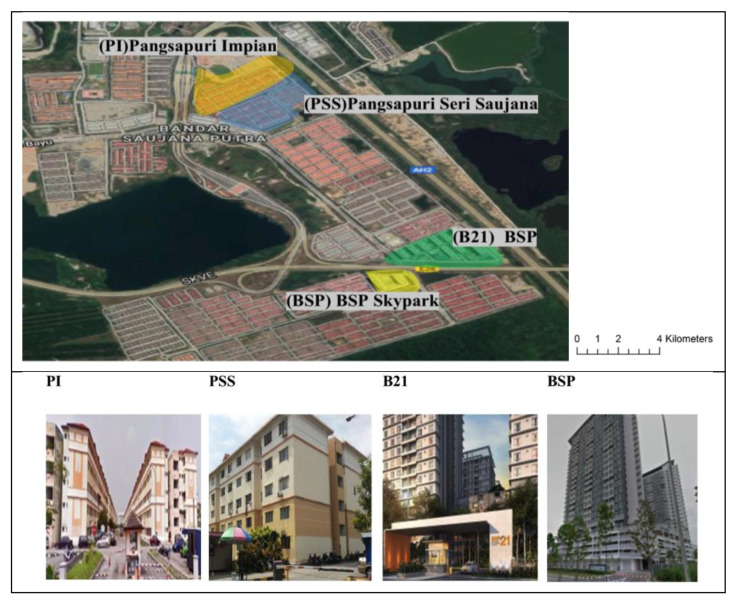
Sampling sites for four different localities of high-rise apartments in Bandar Saujana Putra; (PI) Pangsapuri Impian (orange), (PSS) Pansgapuri Seri Saujana (blue), (B21) BSP 21 (green) and (BSP) BSP Skypark (yellow).

**Figure 2 tropicalmed-05-00067-f002:**
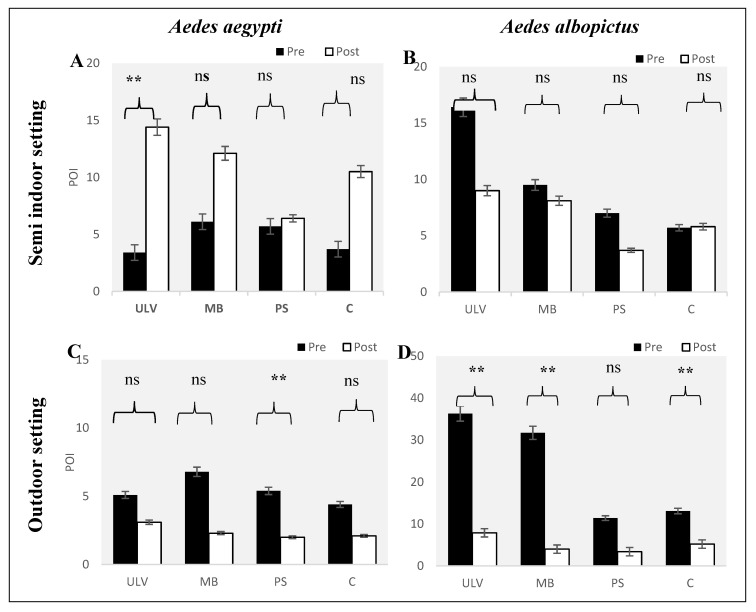
Distribution of pre- and post-treatment based on the positive ovitrap index (POI) of dengue vectors (**A** & **C**: *Aedes aegypti*; **B** & **D**: *Aedes albopictus*) in different environment settings. Note: treatment application of *Bti* (VectoBac^®^) WG are coding with ULV, MB and PS to represent ultra-low volume sprayer, mist blower, pressured sprayer, respectively, and C as the control without any treatment. Mean numbers and standard errors are shown. Significant differences between pre and post-treatments are noted (ns: not significant, ** *p*-value < 0.05).

**Figure 3 tropicalmed-05-00067-f003:**
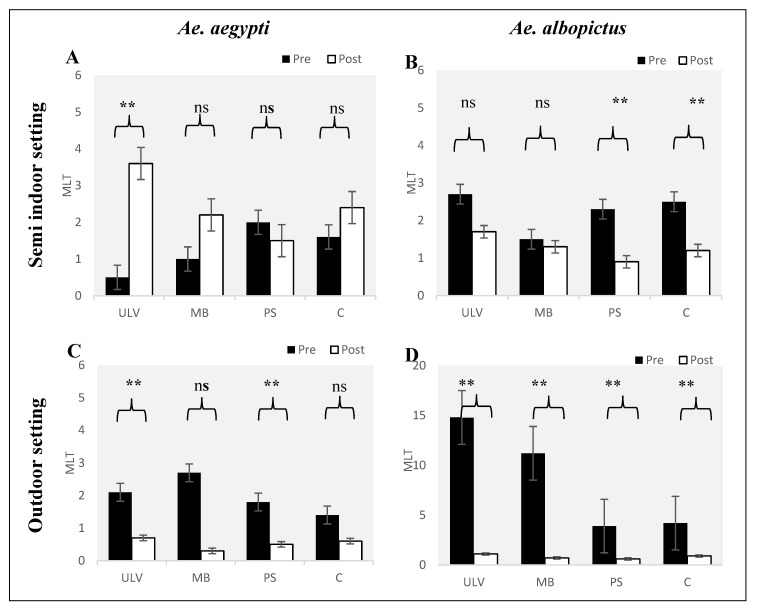
Abundance of pre- and post-treatment based on mean larvae per trap (MLT) of dengue vectors (**A** & **C**: *Aedes aegypti*; **B** & **D**: *Aedes albopictus*) in different environment settings. Note: treatment application of *Bti* (VectoBac^®^) WG are coding with ULV, MB and PS to represent an ultra-low volume sprayer, mist blower, pressured sprayer, respectively, and C as control without any treatment. Mean numbers and standard errors are shown. Significant differences between pre- and post-treatments are noted (ns: not significant, ** *p*-value < 0.05).

**Table 1 tropicalmed-05-00067-t001:** Timeline of the study on the efficacy of *Bti* treatment on *Aedes* sp. using different applications at high-rise buildings.

Locality	Applications	Observational Week for Pretreatment (Grey Color) and Post Treatment (Blue Color)												
		1	2	3	4	5	6	7	8	9	10	11	12	13
Pansapuri Impian (PI)	Ultra-low volume (ULV)	**1st**		**2nd**		**3rd**		**✹**		**✹**		**✹**		
Pangsapuri Seri Saujana (PSS)	Mist blower		**1st**		**2nd**		**3rd**		**★**		**★**		**★**	
BSP21	Pressured sprayer			**1st**		**2nd**		**3rd**		**✚**		**✚**		**✚**
BSP Skypark	Control	**1st**		**2nd**		**3rd**								

**Note:** Baseline study (pre-treatment) based on ovitrap surveillance (grey shading) was conducted for each locality. Each locality is applied with the *Bti* treatment using different applications (blue shading). The ultra-low volume, ULV (**✹**), mist blower (**★**), and pressured sprayer (**✚**) are applied in Pangsapuri Impian (PI), Pangsaputi Seri Saujana (PSS) and BSP 21 (B21), respectively. BSP Skypark was a control locality where no treatment had been applied during the study periods. Post-treatment was conducted a week after completing the treatment application.

**Table 2 tropicalmed-05-00067-t002:** The specification of treatment applications used in this study.

Treatment Application	Types of Area	Recommended Dilution	Equipment Capacity and Capability
ULV	Gutter, dump site, gardens	750 g/12 L	Full tank 35 L can cover up to 300 m^2^. Discharge rate 300 mL/min. There are two nozzle sizes; minimum at 1 hour able to disperse 18 L. Vehicle speed is 5–8 km/h.
Mist blower	Drainage, drain Gutter, dump site, gardens Water reservoir, pond	50 g/12 L 125 g/12 L 250 g/L	Full tank of 12 L can cover up to 50 m^2^. Discharge rate: 120 mL/min There are 6 speeds. Speed level 1 has dispersion distance of 2 m and maximum speed level 6 can disperse to a range of between 12–15 m
Pressured sprayer	Drainage, drain Gutter, dump site, gardens	50 g/12 L 125 g/12L	Full tank of 12 L can cover up to 50 m^2^. Maximum discharge rate: 120 mL/min depends on the pressure given. Light-weight, easy to be carried everywhere. Uses manual hand pump to control the dispersion distance of sprayer.

**Table 3 tropicalmed-05-00067-t003:** Percentage reduction of positive ovitrap index (POI) and mean larvae per trap (MLT) based on species between pre- and post-treatment according to semi indoors and outdoors setting.

**A. Positive Ovitrap Index (POI)**												
**Treatment**	**Semi Indoor (POI)**						**Outdoor (POI)**					
	***Aedes aegypti***			***Aedes albopictus***			***Aedes aegypti***			***Aedes albopictus***		
	**Pre**	**Post**	**%**	**Pre**	**Post**	**%**	**Pre**	**Post**	**%**	**Pre**	**Post**	**%**
ULV	3.4	14.4	+76.4	16.4	9	−45.1	5.1	3.1	−39.2	36.3	7.9	−78.2
Mist blower	6.1	12.1	+49.6	9.5	8.1	−14.7	6.8	2.3	−66.2	31.7	4.0	−87.4
Pressured sprayer	5.7	6.4	+10.9	7.0	3.7	−47.1	5.4	2.0	−63	11.4	3.0	−73.7
Control	3.7	10.5	+64.8	5.7	5.8	+1.7	4.4	2.1	−52.3	13.1	3.4	−74
**B. Mean Larvae per Trap (MLT)**												
**Treatment**	**Semi Indoor (POI)**						**Outdoor (POI)**					
	***Aedes aegypti***			***Aedes albopictus***			***Aedes aegypti***			***Aedes albopictus***		
	**Pre**	**Post**	**%**	**Pre**	**Post**	**%**	**Pre**	**Post**	**%**	**Pre**	**Post**	**%**
ULV	0.5	3.6	86.0	2.7	1.7	−37.0	2.1	0.7	−66.7	14.8	1.1	−92.6
Mist blower	1.0	2.2	54.5	1.5	1.3	−13.3	2.7	0.3	−88.9	11.2	0.7	−93.8
Pressured sprayer	2.0	1.5	25.0	2.3	0.9	−60.9	1.8	0.5	−72.2	3.9	0.6	−84.6
Control	1.6	2.4	33.3	2.5	1.2	−52.0	1.4	0.6	−57.1	4.2	0.9	−78.6

**Note:** Symbol of (−) shows there is an effect on the treatment while (+) shows there is no effect of the treatment on the density of dengue vectors.

**Table 4 tropicalmed-05-00067-t004:** Coefficients result for controlling factor for (**A**) positive ovitrap index (POI) and (**B**) mean larvae per trap (MLT) using hierarchical multiple regressions.

**A. Coefficients Result for Controlling Factor for Positive Ovitrap Index (POI)**					
**Model**	**Unstandardized Coefficients**		**Standardized Coefficients**	**t**	**Significance**
	**B**	**Standard Error**	**Beta**		
Control	−0.035	0.849	−0.013	−0.041	0.968
ULV	−2.378	1.034	−0.885	−2.298	0.051
Mist blower	−2.885	0.358	−1.302	−8.052	0.000
Pressured sprayer	9.459	2.143	1.807	4.413	0.002
**B. Coefficients Result for Controlling Factor for Mean Larvae per Trap (MLT)**					
**Model**	**Unstandardized Coefficients**		**Standardized Coefficients**	**t**	**Significance**
	**B**	**Standard Error**	**Beta**		
Control	−0.081	0.131	−0.191	−0.616	0.552
ULV	−0.270	0.243	−0.638	−1.113	0.298
Mist blower	−0.419	0.084	−1.199	−4.985	0.001
Pressured sprayer	1.065	0.503	1.290	2.118	0.067

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
