# Peer review of "Efficacy of Bacillus thuringiensis Treatment on Aedes Population Using Different Applications at High-Rise Buildings"

_tropicalmed, 2020, doi:10.3390/tropicalmed5020067_

Round 1
Reviewer 1 Report
Dear Authors,
the MS entitled “Efficacy of Bacillus thuringiensis Treatment on Aedes Population Using Different Applications at High-Rise Buildings” is interesting and the aresults appear very useful, on my opinion. nevertheless, some important steps are not described enough in terms of methodology and discussions. I consider the MS suitable to be published only after some changes of major concerns mainly in the methods and in the discussions. Considerations and suggested changes are reported below.
Best regards,
The Reviewer
Major concerns:
1_Methods_ the main lack in this work, on my opinion, is that you not explain why did you used twoindexes POI and MLT, and how. The methodology is not reported and not reported is any reference about. For POI you used ovitraps and this is clear; but how and where did you collect larvae? The larvae were in the same ovitraps used to collect and to count eggs? Or you let the eggs in the ovitraps waiting they to disclose? If so, did you treat also the ovitraps during the treatment? How did you count larvae, in lab? How did you count the eggs to exstimate the mosquito density if yod had to wait their disclosure? All this procedures have to be described or if already used and published you have to report the references.
2_Methods_The chose of the control. You consider the control in you test and this is correct. But you considered 4 buildings, 2 well maintained and 2 in unclean conditions (lines 96-101). If you chose the control belonging to one of this two groups, your control is not impartial. Or you chose another control building with the features of the first group (well maintained) or you can chose a control with features in the middle between the two groups. If you cannot produce this datum at least explain that you are conscious of this limit and try to argue why you were forced to this choice in selecting the control.
3_Methos and discussions_Please explain why you ofter refer to Aedes mosquitoes besides to Aedes albopictus. If you collected other species of Aedes mosquitoes during your study, or you report the species and the relative abundance, or please remove it. Or as alternative, please explain why did you refer to Aedes in general and in this case pleas use “Aedes sp.”
4_Discussions_In the discussions you often repeat the same result as the best technique is the application of bti by mist blower. This is the main result, but you should have to try to discuss widely this datum. For example which could be, on your opinion, the advantages (also economic) from the use of this method respect to the others,.etc..
5_Conclusions_Please explain the sentence “Each method also required different levels of dilution based on the type of the breeding site area.” Moreover, in order to differentiate the Conclusions respect to the discussions, please add some considerations about the intention to carry on this type of study.
Minor concerns:
Line 15-16: “Three different application 16 of Bti treatment were applied at four high-rise buildings in Bandar Saujana Putra.” Three, not four are the tested building, as one is the control, please change.
Line 118-120: Please, specify the formulation of the product based on Bti
Line 138-142: This passage was already reported before.
166-168: This is a method and it should have to be removed from here.
170_172: here, do you refer to cases of dengue? If yes please report this information in terms of cases before and after the treatment. Or do you refer to the density of Aedes aegypti? Please clarify.
231-232: Tab2 caption_ Please explain to which dengue vectors are you referring in this point; which species?
253-255: this passage should have moved to discussion or conclusion, here you still are in the Results section and no opinions or considerations are opportune.
291-303: This passage you already correctly reported in the introduction; please remove it.
306: please specify, adding: “..the results obtained in this study..”
318-323: please, also in this case, this passage is suitable for the introduction, not in discussion.
331-332: This sentence is redundant, it was already reported, please remove it.
336: Which Aedes species?
341-342: This is a repetition of the information, please lighten the discussions..
354: “Source reduction…” please specify reduction of larval breeding sites .
358-359: I suggest to move this as last sentence of this paragraph.
Author Response
Submission of reviewed manuscript (tropicalmed-764962)
I wish to submit a reviewed manuscript for intended publication in Tropical Medicine and Infectious Disease for your consideration.
I have done a necessary correction based in the reviewer comments.
Thank you for your consideration of this manuscript.
Sincerely,
Nazri CD

Reviewer 2 Report
The manuscript titled “Efficacy of Bacillus thuringiensis treatment on Aedes populations using different applications at high-rise buildings” in a semi-indoor and outdoor setting. The manuscript was looking to identify the best application methods of Bti. In general, I am do not have an issue with the study design and the analysis of this manuscript. The major deficiency of the manuscript is the amount of editing that is needed. The manuscript is highly repetitive (two sentences saying the same thing). Another issue with the manuscript (besides the needed English/grammar revisions) is the lack of a discussion. As currently written, I find the discussion to be a restatement of the results without any sort of comparison with previous studies; however, previous studies were acknowledged, just not discuss (what where the results from the previous studies and how do they compare with the studies in this manuscript). My impression from the authors of this manuscript is that Bti will solve the problems of Aedes populations in this location; however, would the local be better served by integrated pest management approach for mosquito control?
Comments
Why is Aedes italicized in the title, but not in any of the rest of the body. Further all scientific names need to be italicized in the body of the manuscript…this should’ve been done prior to submission.
Why were only Aedes investigated? Are there no other mosquito species in this location?
Minor Comments:
Ln 15: application should be plural
Ln 33-34 is really restating the sentence in ln 32; authors should rephrase
Ln 52 needs some attend with regards to grammar and punctuation.
Ln 54: Bacillus thuringensis should be italicized
Ln 77 is odd, why would this paper be the recommendation, would that not be the role of a regulatory agency? I would suggest removing or rephrasing.
Figure 1 is very pixilated. Is it possible to obtain higher resolution photos?
Ln 109-112: two sentences that should be rephrased into 1 (they say the same thing).
LN169: post-treatment should also be hyphenated (especially if pre-treatment is)
LN 171: density of dengue? How was dengue measured, did I miss this in the materials and methods?
Author Response

(The authors gave the same response as above.)

Round 2
Reviewer 1 Report
Dear Authors,
I checked the manuscript after the requested changes. The script could have been improved more than this new version, anyway this version clearer respect to the previous one.
In general I consider the manuscript valid to be published.
Best regards,
The Reviewer
Reviewer 2 Report
comments have been adequately addressed.